# Detection of *Pestalotiopsis abbreviata* sp. nov., the Causal Agent of Pestalotiopsis Leaf Blight on *Camellia japonica* Based on Metagenomic Analysis

**DOI:** 10.3390/jof11080553

**Published:** 2025-07-25

**Authors:** Sung-Eun Cho, Ki Hyeong Park, Keumchul Shin, Dong-Hyeon Lee

**Affiliations:** 1Institute of Agriculture and Life Science, Gyeongsang National University, Jinju 52828, Republic of Korea; secho0324@gmail.com; 2Forest Entomology and Pathology Division, National Institute of Forest Science, Seoul 02455, Republic of Korea; bergpkh0@korea.kr; 3Department of Forest Environmental Resources, Gyeongsang National University, Jinju 52828, Republic of Korea; 4Department of Environment and Forest Resources, Chungnam National University, Daejeon 34134, Republic of Korea

**Keywords:** diagnosis, disease monitoring, leaf spot, metabarcoding

## Abstract

Tree diseases affecting *Camellia japonica* have emerged as a significant threat to the health and longevity of this ornamental tree, particularly in countries where this tree species is widely distributed and cultivated. Among these, *Pestalotiopsis* spp. have been frequently reported and are considered one of the most impactful fungal pathogens, causing leaf blight or leaf spot, in multiple countries. Understanding the etiology and distribution of these diseases is essential for effective management and conservation of *C. japonica* populations. The traditional methods based on pathogen isolation and pure culture cultivation for diagnosis of tree diseases are labor intensive and time-consuming. In addition, the frequent coexistence of the major pathogens with other endophytes within a single *C. japonica* tree, coupled with inconsistent symptom expression and the occurrence of pathogens in asymptomatic hosts, further complicates disease diagnosis. These challenges highlight the urgent need to develop more rapid, accurate, and efficient diagnostic or monitoring tools to improve disease monitoring and management on trees, including *C. japonica*. To address these challenges, we applied a metagenomic approach to screen fungal communities within *C. japonica* trees. This method enabled comprehensive detection and characterization of fungal taxa present in symptomatic and asymptomatic tissues. By analyzing the correlation between fungal dominance and symptom expression, we identified key pathogenic taxa associated with disease manifestation. To validate the metagenomic approach, we employed a combined strategy integrating metagenomic screening and traditional fungal isolation to monitor foliar diseases in *C. japonica.* The correlation between dominant taxa and symptom expression was confirmed. Simultaneously, traditional isolation enabled the identification of a novel species, *Pestalotiopsis,* as the causal agent of leaf spot disease on *C. japonica*. In addition to confirming previously known pathogens, our study led to the discovery and preliminary characterization of a novel fungal taxon with pathogenic potential. Our findings provide critical insights into the fungal community of *C. japonica* and lay the groundwork for developing improved, rapid diagnostic tools for effective disease monitoring and management of tree diseases.

## 1. Introduction

*Pestalotiopsis* (*Pestalotiopsidaceae* and *Amphisphaeriales*) is an ascomycetous genus introduced by Steyaert [1] to accommodate asexual fungi that produce conidia bearing both apical and basal appendages. The genus was typified by *Pestalotiopsis guepinii*, which was first described from isolates obtained from the stems and leaves of *Camellia japonica* in France. *Pestalotiopsis*, a species-rich asexual genus mostly lacking sexual morphs, is widely distributed throughout tropical and temperate regions [2,3,4]. *Pestalotiopsis* exhibits a diverse range of ecological characteristics, functioning as saprophytes and either parasitic or pathogenic fungi. In some cases where the species within this genus are regarded as phytopathogens, they cause a variety of diseases on foliage, stems, and roots, leading to considerable economic losses [4,5,6,7,8], while this group is also commonly isolated as endophytes [4,9,10].

Species of *Pestalotiopsis* can be differentiated by conidial morphology, including conidial length, width, length and color of median cells, and length of the apical appendages from its closely related groups, including *Neopestalotiopsis* and *Pseudopestalotiopsis* [11,12]. However, given the fact that identification based solely on morphological characters proved to be unreliable, phylogenetic species delimitation based on the introduction of multi-locus DNA sequence analyses is currently accepted to assure the identity at the species level [3,4,6].

Recent breakthroughs in bioinformatics, especially metagenomic-based approaches to detect plant pathogens, have revolutionized microbial diagnosis by offering expedited and remarkably sensitive protocols, compared with traditional microbiological approaches including culture-based pathogen isolation protocols [13]. The majority of these culture-independent and high-throughput techniques prioritize assessing entire microbial communities rather than isolating individual or closely related taxa to promptly and effectively detect phytopathogens based on the profile of phytobiome [13,14,15,16].

The genus *Camellia* (*Theaceae*) includes more than 250 plants (“The Plant List”, retrieved from 10th June 2025), and is widely distributed in tropical and subtropical regions including East Asian countries [17]. Of these, *C. japonica* is one of the most popular tree species for essential oil production, ornamental, and medicinal purposes [18]. *Camellia japonica* is a tree species widely distributed and commonly found in the southern regions of South Korea, especially for a variety of ecological and ornamental purposes, including as a hedge plant alongside roads in urban areas in the country.

Due to the popularity of *C. japonica* for various purposes in the country, disease surveillance and monitoring on *C. japonica* were performed during the summer of 2021 and 2022 to implement containment programs for pathogens on the tree. Severely infected leaves of *C. japonica*, showing symptoms including leaf spots and tips and marginal blight upon investigation, were revealed. As part of this study, microbial profiles on the leaves of *C. japonica* were determined using a metagenomic approach to achieve high-throughput identification of a causal agent of leaf spot disease observed on the tree. The occurrence and abundance of the causal pathogen would be higher in the diseased tissues of *C. japonica*. In order to validate the results obtained through metagenomic analysis and to achieve species-level identification of the pathogen, culture-based isolation techniques were additionally applied.

## 2. Materials and Methods

### 2.1. Study Site Description and Sample Collection

The two sampling sites were located in Busan, a city in the southern part of South Korea (35°09′13.4″ N 129°09′03.5″ E, 35°09′08.2″ N 129°09′09.5″ E), which were approximately 100 m apart from each other. The study areas were dominated by *C. japonica*, which were naturally propagated near the sea, with scattered *Pinus thunbergii* trees.

Approximately 10 to 15 leaves per tree of *C. japonica* showing leaf spot symptoms (Figure 1A–C) were randomly sampled, and samples were collected from each of the ten trees at each study site. Samples were placed in individual paper bags and transported to the laboratory for DNA extraction and fungal isolation.

### 2.2. Genomic DNA Extraction and PCR Amplifications, Sequencing

For metagenomic analysis, a total of 24 samples were subjected to DNA extraction. Genomic DNA was extracted from approximately 0.25 g (fresh weight) of each homogenized leaf sample using a DNeasy Plant Mini Kit (Qiagen, Valencia, CA, USA) following the manufacturer’s instructions. The genomic DNAs extracted were quantified using a Nanodrop ND-1000 spectrophotometer (Nanodrop, Wilmington, DE, USA) and adjusted to a final concentration of 20 ng µL^−1^. To obtain a metabarcoding profile on the leaves of *C. japonica*, PCR amplification was achieved using the ITS3 and ITS4 primers [19], targeting the internal transcribed spacer region 2 (ITS2), which is known to have no insertions commonly found in ITS1, producing a 320 bp amplicon in length on average in fungi [20,21]. These were then transported on dry ice to Macrogen Inc. (Seoul, Republic of Korea) for paired-end sequencing using the Illumina MiSeq platform.

For culture-based isolation of the fungal pathogen on *C. japonica*, fungal isolations were directly made to 2% potato-dextrose agar (PDA, Difco Laboratories, Inc., Detroit, MI, USA) supplemented with 100 mg/L^−1^ streptomycin sulphate (Sigma–Aldrich, St. Louis, MO, USA) by transferring spores from the infected leaves of the tree, where, in most cases, conidiomata were abundantly sporulated. Cultures were incubated at 25 °C for three weeks in the dark to allow for sufficient mycelial growth. The mycelium was then scraped from the surface of the agar with sterilized surgical scalpel blades and transferred to 1.5 mL Eppendorf tubes. Genomic DNA was then extracted using a ZR Fungal/Bacterial DNA MiniPrep kit (Zymo Research, Irvine, CA, USA) following the manufacturer’s instructions. The quantity and quality of DNA extracted were evaluated with a spectrophotometer (ND-1000; NanoDrop Technologies, Wilmington, DE, USA) to calibrate the concentration and purity of DNA as PCR templates.

### 2.3. Bioinformatic Processing

Following sequencing, a total of 2,439,406 raw reads were downloaded and processed using QIIME2 version 2022.2 [22]. Denoising was performed with the DADA2 plugin [23] in single-read mode, due to the size of the target amplicon. Sequences were trimmed and quality filtered, with full-length duplicates removed and reads sorted by abundance. For ITS sequences, denoised reads were used to infer amplicon sequence variants (ASV), and chimeric sequences were removed using the UNITE database as a reference [24], which served as the basic units of observation. Taxonomic classification was performed using a classifier trained with the scikit-learn library, based on the UNITE database (release 8.0). For classification, only ASV with a minimum of 10 sequence reads were included. The final processed ASV table comprised 1,166,426 reads across 1704 ASV.

### 2.4. Data Preparation and Metagenomic Statistical Analysis

Fungal diversity profiling was conducted to evaluate the effects of one factor—diseased vs. healthy. Subsequently, the dataset processed from QIIME2 was imported into R and converted into a *phyloseq* object for further analyses. Taxonomic composition, diversity metrics, and ordination analyses were subsequently performed to facilitate interactive visualization and statistical interpretation.

Permutational multivariate analysis of variance (PERMANOVA) was performed to assess if significant differences (*p* < 0.05) existed between healthy and diseased trees of *C. japonica* in the relative abundance of fungal microbiota, based on the number of ASV, in relation to the factors under study.

PERMANOVA analyses using the ‘adonis2() function’ from the vegan package were performed based on a Bray–Curtis similarity matrix [25]. When the number of unique permutations was fewer than 15, Monte Carlo permutation *p*-values were reported. In cases where significant interaction effects were observed, a posteriori pairwise comparisons were conducted using 9999 permutations under a reduced model.

Subsequently, differential abundance testing was performed on the top 10 most abundant taxa to identify statistically significant differences in their relative abundances across samples through PERMANOVA analysis based on the ASV dataset.

These were all achieved using the *microViz* package (version 0.12.7) [26], implemented in R version 3.6.1. ASV occurring fewer than two times were removed from the dataset, and features present in only one sample were excluded. Unless otherwise specified, all procedures were performed using default settings.

### 2.5. Microscopy

Fungal structures were mounted on microscope slides in water and later replaced with 85% lactic acid for further observation. The structures were examined using a Zeiss AX10 Imager A2 (Carl Zeiss Microscopy GmbH, Göttingen, Germany) equipped with an Axiocam 506 color digital camera. At least 30 measurements were taken for taxonomically relevant structures when possible.

### 2.6. Multi-Gene Phylogenetic Analyses

To confirm the result obtained from the metagenomic analysis, the culture-based isolation approaches were employed to ensure the identity of the fungal pathogen isolated from *C. japonica* at the species level.

Two isolates, GNUFP284 and GNUFP285, representing each sampling location, obtained in this study, were subjected to sequencing and phylogenetic analyses based on three gene regions. These included the internal transcribed spacer region (ITS), partial β-tubulin (*tub2*), and partial translation elongation factor 1-alpha (*tef1*), which were amplified using the primers ITS5/ITS4 [19], T1/Bt-2b [27,28], and EF1-728F/EF-2 [29,30], respectively. Amplification conditions for ITS and *TEF* followed Crous et al. [5] and for *TUB*, Lee et al. [31]. The resulting PCR products were submitted to Macrogen (Seoul, Korea) for forward and reverse sequencing reactions.

The sequences of *Pestalotiopsis* spp. were retrieved from GenBank (Table 1). Sequences for each of the three gene regions, ITS, *TUB*, and *TEF*, were then aligned using the online interface of MAFFT version 7 (http://mafft.cbrc.jp/alignment/server, accessed on 29 May 2025) [32], with the iterative refinement method (FFT-NS-i settings) selected. Sequence alignments were manually edited in MEGA7 [33], producing a concatenated dataset of the three gene regions.

Two different phylogenetic analyses were employed, including maximum parsimony (MP) analyses using MEGA7 and maximum likelihood (ML) tests using RAxML HPC BlackBox version 8.1.11 [34,35], using the default option with the GTR substitution model implemented in the CIPRES cluster server (https://www.phylo.org/, accessed on 31 May 2025) at the San Diego Supercomputing Center. For both MP and ML analyses, *Pestalotiopsis jesteri* was used as the outgroup taxon.

## 3. Results

### 3.1. Metagenomic Assessment of Fungal Diversity on Camellia japonica

Deep sequencing of fungal-targeted ITS1 amplicons yielded 1,166,426 high-quality reads, from which 1704 unique amplicon sequence variants (ASV) were inferred. For downstream analysis, a subset of 961 ASV assigned to the phylum *Ascomycota* was extracted from the processed dataset and used for further analyses, and an overview of the fungal community composition is presented in Table 2.

The dataset was subsequently filtered using the tax_filter() function with default parameters from the *microViz* package (v0.12.7) [26] in R, in order to retain the most abundant taxa associated with *C. japonica.* This consequently resulted in a total of 90 taxa with a relative abundance exceeding 0.10%.

### 3.2. Detection of a Putative Fungal Pathogen Associated with Leaf Blight on Camellia japonica Based on Metagenomic Analysis

Within the phylum *Ascomycota* on *C. japonica*, the most abundant genus was *Pestalotiopsis* with a total abundance of 31.8%. This tendency was most often observed regardless of the sampling location (Figure 2).

In addition to *Pestalotiopsis* sp., several other putative fungal pathogens, including *Aureobasidium* sp. and *Cladosporium* sp., etc., that might cause leaf spot disease on *C. japonica*, were observed. However, these taxa can likely be excluded based on the relatively lower abundance than that observed from *Pestalotiopsis* sp., as well as the characteristic symptoms and signs observed on the leaf surfaces of the affected trees (Figure 2).

Among the 10 most prevalent fungal taxa observed from *C. japonica*, differential abundance testing was conducted based on PERMANOVA analysis, and it was clearly shown that *Pestalotiopsis* sp. was identified as the predominant taxon, showing statistically significant differences in its relative abundance between healthy and diseased *C. japonica* trees (Figure 3) and within the diseased *C. japonica* trees (Figure 4).

To assess the pathogenicity of *Pestalotiopsis* sp. that was identified through metagenomic analysis, inoculation trials were conducted using a single isolate (GNUFP284). A conidial suspension (1 × 10^5^ conidia/mL) was sprayed onto the leaves of *C. japonica* seedlings, while control plants were treated with sterile distilled water. Two weeks after inoculation, characteristic leaf spot symptoms developed on the inoculated seedlings, and *Pestalotiopsis* sp. was successfully re-isolated from the lesions, thereby fulfilling Koch’s postulates.

### 3.3. Identification of a Putative Fungal Pathogen Associated with Leaf Blight on Camellia japonica

#### 3.3.1. Multi-Gene Phylogenetic Analyses

Three gene regions, including ITS, *tub2*, and *tef1*, were successfully sequenced and deposited in GenBank under accession numbers OR478187–188 (ITS), OR480768–769 (*tub2*), and OR480770–771 (*tef1*). The sequences obtained from each gene region were aligned with those of closely related *Pestalotiopsis* species, as identified through BLAST searches (version BLAST+ 2.16.0) in the NCBI nucleotide database. Phylogenetic analyses based on the concatenated sequences of ITS, *tub2*, and *tef1* yielded a tree (Figure 5). Although minor differences were observed between the topologies generated by the maximum likelihood (ML) and maximum parsimony (MP) methods, both analyses consistently supported that the isolates from *C. japonica* represent a previously undescribed species. This novel species is closely related to *P. portugalica* but is phylogenetically distinct from it and all other described *Pestalotiopsis* species.

#### 3.3.2. Taxonomy

Morphological comparisons, including conidiomata and conidia, and phylogenetic inference based on three gene regions provided sufficient evidence that the *Pestalotiopsis abbreviata* sp. nov. isolated from *Camellia japonica* represents an undescribed species. This species is described as follows:

***Pestalotiopsis abbreviata*** D.Hyeon Lee & S.E. Cho, ***sp. nov.*** MB 859637. Figure 6.

*Etymology*: The epithet refers to the reduced or absent basal appendages observed in this species.

*Typus*: **Republic of Korea**, Busan, Haeundae-gu, U-dong, 708-7, 35°09′13.4″ N, 129°09′03.5″ E, on leaves of *C. japonica*, 24 June 2021. *D.H. Lee* (**holotype** KACC 410709, ex-holotype culture GNUFP284).

*Description*: *Conidiomata* developed on PDA, pycnidial, globose to clavate, appearing aggregated or scattered, partially embedded to erumpent, dark brown to black, and measuring up to 550 μm in diameter; they released dark brown to black, globose conidial masses. *Conidiophores* often absent or reduced to conidiogenous cells; when present, septate near the base, unbranched or basally branched, subcylindrical, and hyaline. *Conidiogenous cells* discrete or integrated, ampulliform to clavate or subcylindrical, hyaline, smooth-walled, broad-based, and measure 7–35 × 3–5 μm. *Conidia* fusoid to ellipsoid, straight to slightly curved, 4-septate, and slightly constricted at the septa, measuring 13–18 × 4.5–7.1 μm. *Basal cell* hyaline, conic to obconic with a truncate base, thin-walled and verruculose, 3–6 μm long. *Median cells* doliiform, pale brown to brown with darker septa, totaling 11–14.5 μm. *Apical cell* hyaline, obconic, thin-walled, and verruculose, 3.0–5.0 μm long. *Apical appendages* one to two unbranched (mostly one), filiform and arise from the apex, 9–23 μm long. *Centric basal appendage* absent. *Sexual states* not observed.

*Culture characteristics*: Colonies, on 2% PDA, white, with flocculent aerial mycelium, which grow relatively fast, reaching the edge of the Petri dish 7 to 10 days after incubation at 25 °C in darkness. Scattered black conidiomata formed on 2% PDA.

*Additional specimen examined*: **Republic of Korea,** Busan, Haeundae-gu, U-dong, 710-1, 35°09′08.2″ N 129°09′09.5″ E, on leaves of *C. japonica*, 10 Sep. 2022. *D.H. Lee* culture GNUFP285.

*Notes*: *Pestalotiopsis abbreviata* is distinguished *from P. portugalica* by several distinct morphological characters. This includes non-proliferating, ampulliform to clavate or subcylindrical conidiogenous cells lacking a collarette, while *P. portugalica* shows percurrently proliferating conidiogenous cells (2–6 times) with a distinct collarette and periclinal thickening. The apical appendages in *P. abbreviata* are unbranched (mostly one) and arise directly from the apex, whereas those in *P. portugalica* are 1–3, often branched, and arise from an apical crest. The basal appendage is consistently absent in *P. abbreviata* but occasionally present in *P. portugalica*. In addition, the conidiomata of *P. abbreviata* are larger (up to 550 μm), aggregated, and erumpent, compared with the smaller (200–400 μm), solitary, and semi-immersed conidiomata of *P. portugalica* [4].

## 4. Discussion

The application of advanced molecular approaches—particularly microbial profiling targeting rRNA DNA regions based on metagenomics in plant pathology and applied microbiology—has become essential for the accurate and rapid detection of plant pathogens. This is particularly important given the increasing emergence of virulent strains, which drive the prevalence and severity of plant diseases [16]. This study evaluated the potential application of metagenomic analysis for effective monitoring of tree diseases, particularly for leaf blight disease caused by *Petalotiopsis* spp. on *Camellia japonica,* which is the tree disease that has been reported on *C. japonica* from several countries [36,37]. To validate the reliability of this approach, metagenomic profiling was performed in parallel with conventional culture-based isolation and identification of pathogens. In comparing the two methods, this study aimed to assess the feasibility of employing metagenomics as a complementary tool for disease surveillance and diagnosis.

In recent years, metagenomic approaches have emerged as powerful tools for investigating plant-associated microbial communities, offering high-resolution insights into pathogen diversity and community dynamics without the limitations of culture-based methods. This is partly due to the fact that it is associated with increasing accessibility, allowed either through the lower cost of these methods or the simplification of the processes involved. Despite its limitations in resolving certain genera with low interspecific variability—such as *Aspergillus*, *Cladosporium*, *Fusarium*, *Penicillium*, and *Trichoderma*, which exhibit narrow or absent barcode gaps within the internal transcribed spacer (ITS) regions—ITS remain the primary and most widely accepted molecular markers for fungal identification [38].

In this study, the application of metagenomic analysis targeting the ITS region enabled the detection and taxonomic assignment of *Pestalotiopsis* sp. within the fungal community of *C. japonica*. Although additional loci such as *tub2* or *tef1* may offer higher resolution for species-level identification within the genus *Pestalotiopsis*, the ITS regions remain the most widely used DNA barcode for fungal taxonomy, including for the members of this genus [38]. In the present study, ITS-based metagenomic analysis proved sufficient to reveal the clear predominance of *Pestalotiopsis* species in symptomatic *C. japonica* tissues, thereby supporting its role as a primary pathogen associated with leaf blight symptoms in this host. These findings demonstrate the effectiveness of ITS-based metagenomic profiling for identifying key pathogenic taxa within complex plant-associated microbiomes.

As expected, a diverse group of fungi associated with *C. japonica* was detected in this study. Among the top ten most dominant fungal taxa identified in the leaves of *C. japonica*, several genera were consistently abundant, including both commonly reported endophytic or pathogenic taxa—such as *Pestalotiopsis*, *Cladosporium*, and *Alternaria*—as well as less frequently observed genera such as *Epicoccum*, *Didymella*, *Nigrospora*, *Colletotrichum*, *Fusarium*, *Phoma*, and *Aspergillus*. Notably, the prominence of *Pestalotiopsis* in these metagenomic data was consistent with results from culture-based isolation, as well as with the characteristic morphological features of this genus observed on symptomatic leaf tissues. Together, these findings provide strong evidence supporting its role as a causal pathogen of leaf blight in *C. japonica*.

*Pestalotiopsis* species, including *P. portugalica* and *P. camelliicola,* have been reported as causal agents of leaf spot disease on *Camellia* spp. in China [36] and the Republic of Korea [39]. Additionally, our study confirmed the presence of a novel fungal species within the genus *Pestalotiopsis*, identified as a causal agent of leaf spot disease on *Camellia* spp. This newly described pathogen, designated as *P. abbreviata*, expands the current understanding of the diversity and pathogenicity of *Pestalotiopsis* species associated with *Camellia* hosts. Although phylogenetic analysis revealed *P. portugalica* as the closest relative of the novel species, clear morphological differences were observed between them. Moreover, both maximum parsimony (MP) and maximum likelihood (ML) analyses distinctly supported the classification of this novel species as a separate taxon. In addition, comparative evaluation with *P. cameliicola*, previously reported from *C. japonica* in China [36], demonstrated notable differences in conidial size and other morphological characteristics, further substantiating its distinction as a new species. These findings highlight the need for further investigation into the epidemiology and management of leaf spot diseases caused by this emerging pathogen in the country.

## 5. Conclusions

This study evaluated the applicability of metagenomic analysis as a tool for effective disease diagnosis and monitoring in *Camellia japonica*, a widely cultivated ornamental tree increasingly affected by foliar diseases such as leaf blight and leaf spot. By combining high-throughput sequencing of the ITS region with conventional culture-based isolation, we comprehensively characterized the fungal communities associated with symptomatic and asymptomatic tissues. The clear dominance of *Pestalotiopsis* species in symptomatic samples strongly supports their role as primary pathogens responsible for disease manifestation in *C. japonica*. Furthermore, traditional isolation methods enabled the identification and pathogenicity validation of a previously undetermined species of *Pestalotiopsis*, reinforcing the reliability of the metagenomic approach. These findings demonstrate that ITS-based metagenomic profiling can effectively complement classical diagnostic methods by enhancing detection speed, accuracy, and resolution in complex plant microbiomes. The integration of both approaches provides a robust framework for early pathogen detection, disease surveillance, and the development of targeted management strategies. Overall, this study underscores the practical and scientific value of metagenomic tools in forest pathology and highlights their potential for improving disease management systems in ornamental and ecologically significant tree species.

## Figures and Tables

**Figure 1 jof-11-00553-f001:**
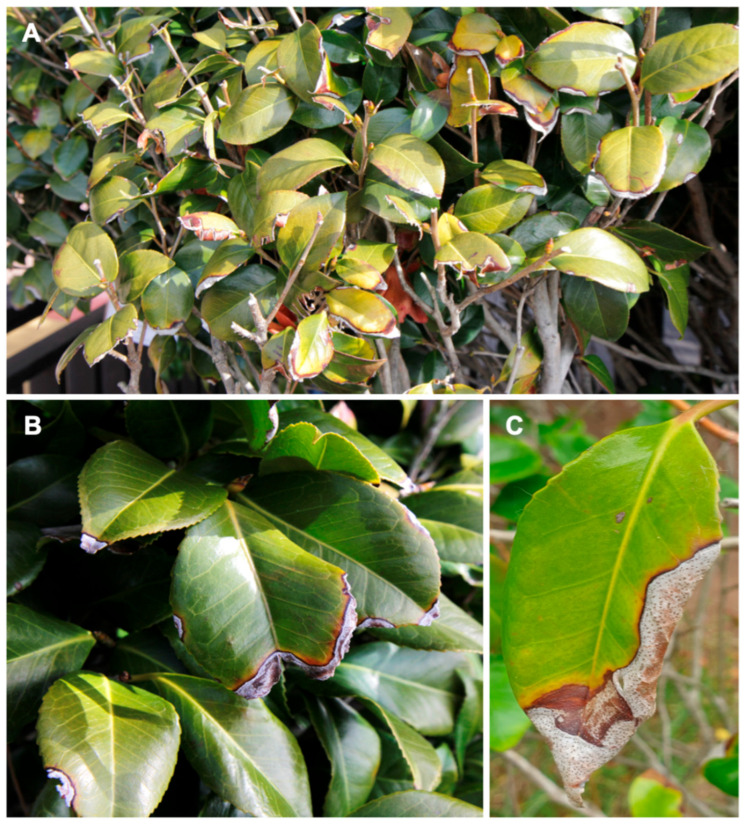
Symptoms of leaf spot and leaf blight on *Camellia japonica*. (**A**) Landscape view of diseased leaves of *C. japonica*. (**B**,**C**) Close-up view of diseased leaves of *C. japonica*.

**Figure 2 jof-11-00553-f002:**
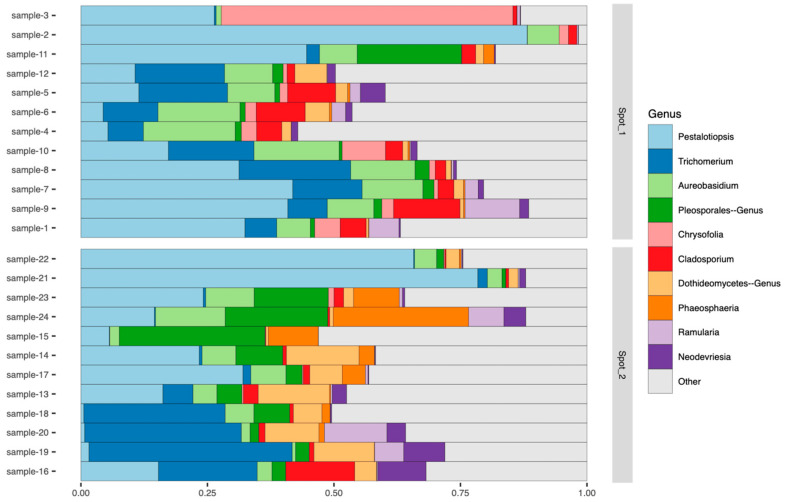
The 10 most abundant taxa within the phylum Ascomycota observed on *C. japonica*.

**Figure 3 jof-11-00553-f003:**
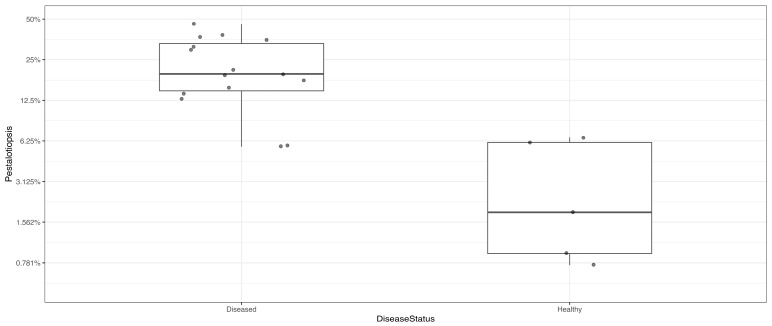
Significant differences in the dominance of *Pestalotiopsis* sp. in its relative abundance between healthy and diseased *C. japonica* trees.

**Figure 4 jof-11-00553-f004:**
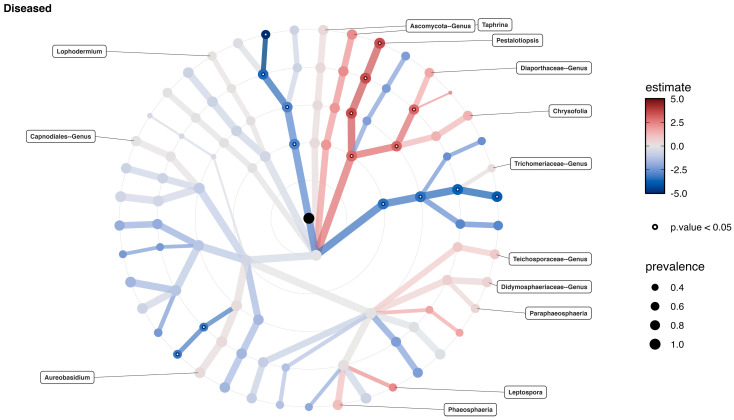
Significant differences in the dominance of *Pestalotiopsis* sp. in its relative abundance within the diseased *C. japonica* trees. Each circle (node) represents a taxonomic unit from phylum to genus level. Red branches indicate taxa significantly abundant in diseased samples, while blue branches indicate taxa that are less abundant. The intensity of the color corresponds to the LDA score (estimate), and the size of each dot reflects the relative prevalence of the taxon across samples. Nodes with a black border denote statistical significance (*p* < 0.05).

**Figure 5 jof-11-00553-f005:**
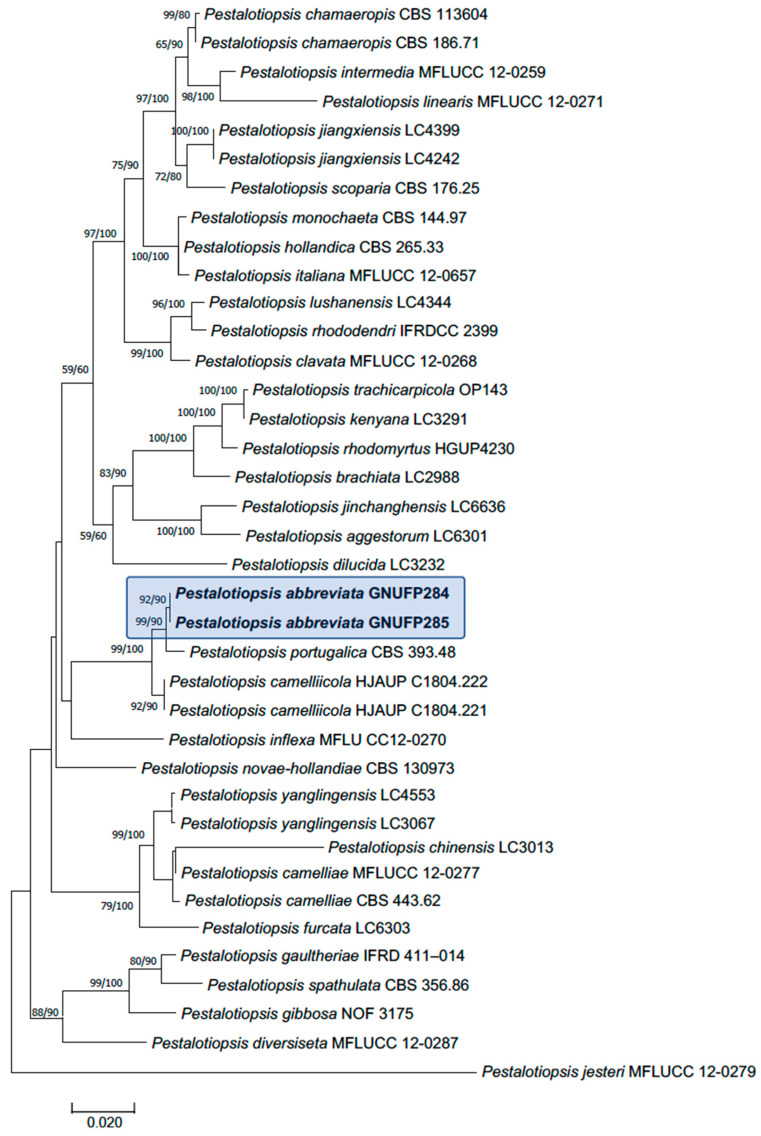
Phylogenetic trees based on maximum likelihood (ML) analysis of a combined dataset of ITS, *tub2*, and *tef1* gene sequences for *Pestalotiopsis* species. Isolates in bold and highlighted are the new species of *Pestalotiopsis* described in this study. Bootstrap values > 50% for MP and maximum likelihood (ML) are presented above branches as ML/MP. The scale bar indicates 0.02 changes.

**Figure 6 jof-11-00553-f006:**
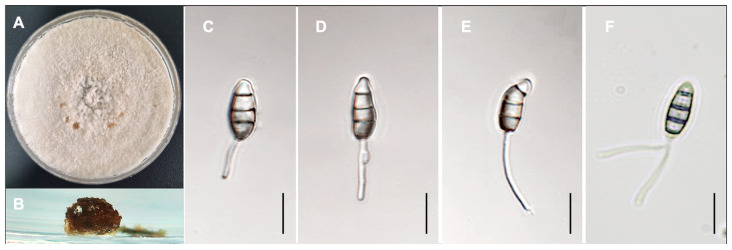
*Pestalotiopsis abbreviate* sp. nov. (**A**) Colony on PDA after 10 days at 25 °C. (**B**) Conidiomata, (**C**–**F**) Conidia. Scale bars: 10 µm (**C**–**F**).

**Table 1 jof-11-00553-t001:** Isolates and GenBank accession numbers used for phylogenetic analyses in this study.

Species	Isolates	ITS	TUB	TEF
*P. abbreviata*	GNUFP284	OR478187	OR480768	OR480770
*P. abbreviata*	GNUFP285	OR478188	OR480769	OR480771
*P. aggestorum*	LC6301	KX895015	KX895348	KX895234
*P. brachiata*	LC2988	KX894933	KX895265	KX895150
*P. camelliae*	CBS 433.62	KM199336	KM199424	KM199512
*P. camelliae*	MFLUCC 12-0277	JX399010	JX399041	JX399074
*P. camelliicola*	HJAUP C1804.221	PP962357	PP952229	PP952236
*P. camelliicola*	HJAUP C1804.222	PP962358	PP952230	PP952235
*P. chamaeropis*	CBS 113604	KM199323	KM199389	KM199471
*P. chamaeropis*	CBS 186.71	KM199326	KM199391	KM199473
*P. chinensis*	LC3013	KX894939	KX895271	KX895156
*P. clavata*	MFLUCC 12-0268	JX398990	JX399025	JX399056
*P. dilucida*	LC3232	KX894961	KX895293	KX895178
*P. diversiseta*	MFLUCC 12-0287	NR 120187	JX399040	JX399073
*P. furcata*	LC6303	KX895016	KX895349	KX895235
*P. gaultheriae*	IFRD 411–014	KC537805	KC537819	KC537812
*P. gibbosa*	NOF 3175	LC311589	LC311590	LC311591
*P. hollandica*	CBS 265.33	KM199328	KM199388	KM199481
*P. inflexa*	MFLUCC 12-0270	JX399008	JX399039	JX399072
*P. intermedia*	MFLUCC 12-0259	JX398993	JX399028	JX399059
*P. italiana*	MFLUCC 12-0657	KP781878	KP781882	KP781881
*P. jesteri*	MFLUCC 12-0279	JX399012	JX399043	JX399076
*P. jiangxiensis*	LC4242	KX895035	KX895327	KX895213
*P. jiangxiensis*	LC4399	KX895009	KX895341	KX895227
*P. jinchanghensis*	LC6636	KX895028	KX895361	KX895247
*P. kenyana*	LC3291	KX894962	KX895294	KX895179
*P. linearis*	MFLUCC 12-0271	JX398992	JX399027	JX399058
*P. lushanensis*	LC4344	KX895005	KX895337	KX895223
*P. monochaeta*	CBS 144.97	KM199327	KM199386	KM199479
*P. novae-hollandiae*	CBS 130973	NR_147557	KM199425	KM199511
*P. portugalica*	CBS 393.48	KM199335	KM199422	KM199510
*P. rhododendri*	IFRDCC 2399	KC537804	KC537818	KC537811
*P. rhodomyrtus*	HGUP4230	KF412648	KF412642	KF412645
*P. scoparia*	CBS 176.25	KM199330	KM199393	KM199478
*P. spathulata*	CBS 356.86	NR 147558	KM199423	KM199513
*P. trachicarpicola*	OP143	KC537809	KC537823	KC537816
*P. yanglingensis*	LC3067	KX894949	KX895281	KX895166
*P. yanglingensis*	LC4553	KX895012	KX895345	KX895231

**Table 2 jof-11-00553-t002:** Overview of fungal microbiota composition on *Camellia japonica*.

	Class	Order	Family	Genus
Classified ASV	876	798	692	626
Identified taxa	11	47	119	180

## Data Availability

The original contributions presented in this study are included in the article. The raw sequence data generated through metagenomic sequencing have been deposited in the NCBI Sequence Read Archive (SRA) under the BioProject accession number PRJNA1293874. Further inquiries can be directed to the corresponding authors.

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
