# Peer review of "Detection of *Pestalotiopsis abbreviata* sp. nov., the Causal Agent of Pestalotiopsis Leaf Blight on *Camellia japonica* Based on Metagenomic Analysis"

_jof, 2025, doi:10.3390/jof11080553_

Round 1

Reviewer 1 Report

This study systematically elucidated the fungal community associated with leaf spot disease in Camellia japonica through a combined approach employing metagenomic analysis and conventional isolation techniques. The identification and characterization of the novel species Pestalotiopsis abbreviata demonstrates significant novelty and application value. The research design is sound, integrating advanced methodologies with traditional validation, thereby providing a valuable framework for the rapid diagnosis of tree diseases. However, further refinement of linguistic precision is recommended to enhance the manuscript's rigor and readability.

  1. Lines 19-20: Subject-verb disagreement. Change "is" to "are" to correct the conflict between a plural subject and a singular verb. It should be: "Pestalotiopsis spp. have been frequently reported and are considered one of the most impactful diseases."
  2. Line 23: Change "on" to "of". It should be: "diagnosis of tree diseases."
  3. Line 49: Subject-verb disagreement. Change the verb to the singular form "exhibits".
  4. The reference materials in lines 68-70 are missing. Suggested references “CaSun1, a SUN family protein, governs the pathogenicity of Colletotrichum camelliae by recruiting CaAtg8 to promote mitophagy”
  5. Line 89: ("apart each other") Add the preposition "from". Change to "apart from each other".
  6. Line 90: Subject-verb disagreement. Change "is" to "were".
  7. Line 92: Noun number disagreement. Change "symptom" to the plural form "symptoms of leaf spots".
  8. Line 95: Change “DNA and fungal isolation” to “DNA extraction and fungal isolation".
  9. Line 102: Subject-verb disagreement & noun number. Change "was" to "were" and "extractions" to "extraction".
  10. Line 125: Change the plural form "processings" to the singular "Processing".
  11. Lines 169-170: The phrase "retained in this study" is a redundant expression.
  12. Line 186: ("jestri,") Remove the comma after "jestri".

This study systematically elucidated the fungal community associated with leaf spot disease in Camellia japonica through a combined approach employing metagenomic analysis and conventional isolation techniques. The identification and characterization of the novel species Pestalotiopsis abbreviata demonstrates significant novelty and application value. The research design is sound, integrating advanced methodologies with traditional validation, thereby providing a valuable framework for the rapid diagnosis of tree diseases. However, further refinement of linguistic precision is recommended to enhance the manuscript's rigor and readability.

  1. Lines 19-20: Subject-verb disagreement. Change "is" to "are" to correct the conflict between a plural subject and a singular verb. It should be: "Pestalotiopsis spp. have been frequently reported and are considered one of the most impactful diseases."
  2. Line 23: Change "on" to "of". It should be: "diagnosis of tree diseases."
  3. Line 49: Subject-verb disagreement. Change the verb to the singular form "exhibits".
  4. The reference materials in lines 68-70 are missing. Suggested references “CaSun1, a SUN family protein, governs the pathogenicity of Colletotrichum camelliae by recruiting CaAtg8 to promote mitophagy”
  5. Line 89: ("apart each other") Add the preposition "from". Change to "apart from each other".
  6. Line 90: Subject-verb disagreement. Change "is" to "were".
  7. Line 92: Noun number disagreement. Change "symptom" to the plural form "symptoms of leaf spots".
  8. Line 95: Change “DNA and fungal isolation” to “DNA extraction and fungal isolation".
  9. Line 102: Subject-verb disagreement & noun number. Change "was" to "were" and "extractions" to "extraction".
  10. Line 125: Change the plural form "processings" to the singular "Processing".
  11. Lines 169-170: The phrase "retained in this study" is a redundant expression.
  12. Line 186: ("jestri,") Remove the comma after "jestri".

Author Response

This study systematically elucidated the fungal community associated with leaf spot disease in Camellia japonica through a combined approach employing metagenomic analysis and conventional isolation techniques. The identification and characterization of the novel species Pestalotiopsis abbreviata demonstrates significant novelty and application value. The research design is sound, integrating advanced methodologies with traditional validation, thereby providing a valuable framework for the rapid diagnosis of tree diseases. However, further refinement of linguistic precision is recommended to enhance the manuscript's rigor and readability.

  1. Lines 19-20: Subject-verb disagreement. Change "is" to "are" to correct the conflict between a plural subject and a singular verb. It should be: "Pestalotiopsis spp. have been frequently reported and are considered one of the most impactful diseases."
  2. Line 23: Change "on" to "of". It should be: "diagnosis of tree diseases."
  3. Line 49: Subject-verb disagreement. Change the verb to the singular form "exhibits".
  4. Line 186: ("jestri,") Remove the comma after "jestri".
  5. Line 89: ("apart each other") Add the preposition "from". Change to "apart from each other".
  6. Line 90: Subject-verb disagreement. Change "is" to "were".
  7. Line 92: Noun number disagreement. Change "symptom" to the plural form "symptoms of leaf spots".
  8. Line 95: Change “DNA and fungal isolation” to “DNA extraction and fungal isolation".
  9. Line 102: Subject-verb disagreement & noun number. Change "was" to "were" and "extractions" to "extraction".
  10. Line 125: Change the plural form "processings" to the singular "Processing".

Response : We appreciate the reviewer’s insightful comments above; to take suggestions into account, we have changed them in the manuscript accordingly.

  1. The reference materials in lines 68-70 are missing. Suggested references “CaSun1, a SUN family protein, governs the pathogenicity of Colletotrichum camelliae by recruiting CaAtg8 to promote mitophagy”

Response : We appreciate this observation. In this regard, we have added the following reference to the manuscript as suggested.

Zhao, D.W.; Hodkinson, T.R.; Parnell, J.A. Phylogenetics of global Camellia (Theaceae) based on three nuclear regions and its implications for systematics and evolutionary history. J. Syst. Evol. 2023, 61, 356-368.

  1. Lines 169-170: The phrase "retained in this study" is a redundant expression.

Response: We agree with the reviewer and have rephased it to ‘Obtained’

Reviewer 2 Report

Just need to some improvements  

I have some comments, as follows:

  1. Line 81-85:"It is assumed" to end:

Delete it or remove to another part

  1. Line 99-100:"D, Colony on

PDA after 10 days at 25°C. ; E-I, Conidia. Bars, 10 um (E to I)":

These photos remove them to results

  1. "2.4. Data preparation and statistical analysis":

This section remove it to last section in material and methods

  1. "Table 1. Isolates and GenBank accession numbers used for phylogenetic analyses in this study.":

If this is your results in this study , you must remove it to results

  1. "3.3.2. Taxonomy":

The authors must add photos to show the characteristics

  1. Line 251:"Morphological comparisons":

The authors must write in materials and methods what are the morphological characteristics used to identify this species

  1. Line 255: "(Figure 1)":

Separate these photos to be for section 3.3.2

  1. The authors must add a separate section for conclusions without repeating the results and discussion

Author Response

I have some comments, as follows:

  1. Line 81-85:"It is assumed" to end:

Delete it or remove to another part

Response: We appreciate the reviewer’s thoughtful comment. The sentence in question was intended to present the hypothesis and objective of the study. Rather than removing or relocating it, we have revised the sentence to be more concise and clear, while retaining its original purpose.

  1. Line 99-100:"D, Colony on

PDA after 10 days at 25°C. ; E-I, Conidia. Bars, 10 um (E to I)":

These photos remove them to results

  1. Line 255: "(Figure 1)":

Separate these photos to be for section 3.3.2

 Response: Thank you for your comments. To take your suggestion into account, the figures have been separated into Figure 1 and Figure 6, and these were rearranged appropriately within the manuscript.

  1. "2.4. Data preparation and statistical analysis":

This section remove it to last section in material and methods

Response: We appreciate the reviewer’s valuable comment. As the experimental design aimed to demonstrate a direct association between the estimated pathogen density (as inferred by metagenomic analysis) and disease occurrence, we further conducted culture-based isolation to confirm the identity of the dominant pathogen detected in the metagenomic data. Therefore, we believe that structuring the manuscript according to each experimental section—metagenomic analysis followed by culture-based validation—is appropriate and scientifically sound. We hope the reviewer agrees with this approach. Thank you once again for your insightful feedback. 

  1. "Table 1. Isolates and GenBank accession numbers used for phylogenetic analyses in this study.":

If this is your results in this study , you must remove it to results

Response: We appreciate this observation. Some of the data were obtained from the NCBI GenBank and are not exclusively generated in this study; therefore, they were included in the Materials and Methods section. We kindly ask for your understanding regarding this decision

  1. "3.3.2. Taxonomy":

The authors must add photos to show the characteristics

Response: We appreciate the valuable suggestion and we have added the photo to the figure accordingly as suggested.

  1. Line 251:"Morphological comparisons":

The authors must write in materials and methods what are the morphological characteristics used to identify this species

Response: We appreciate your comments and we have added the additional information regarding the morphological characteristics used to identify to the manuscript (Line 251) as suggested.

Reviewer 3 Report

The work is novel, makes appropriate use of complementary methods and represents a contribution to the related field of research. However, the English could be improved, particularly in the introduction section; the rest is adequate.

The English could be improved, particularly in the introduction section. Example: The ascomycete genus Pestalotiopsis (Pestalotiopsidaceae, Amphisphaeriales) established by Steyaert [1] to accommodate an asexual fungus having appendage bearing conidia isolated from stems and leaves of Camellia japonica collected in France with the
type species of Pestalotiopsis guepinii. This sentence is quite extense and lacks sense.

Author Response

The English could be improved, particularly in the introduction section. Example: The ascomycete genus Pestalotiopsis (Pestalotiopsidaceae, Amphisphaeriales) established by Steyaert [1] to accommodate an asexual fungus having appendage bearing conidia isolated from stems and leaves of Camellia japonica collected in France with the
type species of Pestalotiopsis guepinii. This sentence is quite extense and lacks sense.

Response : We appreciate the reviewer’s insightful comment above; to take suggestions into account, we have rephrased the sentence in the manuscript accordingly.

Pestalotiopsis (Pestalotiopsidaceae, Amphisphaeriales) is an ascomycetous genus introduced by Steyaert [1] to accommodate asexual fungi that produce conidia bearing both apical and basal appendages. The genus was typified by Pestalotiopsis guepinii, which was first described from isolates obtained from the stems and leaves of Camellia japonica in France.

Reviewer 4 Report

So far as I know, the use of metagenomic analysis for identification of a disease causing organism is unique.  

Editing: line 97,  move per tree to right after leaves
line 144: 'if ' or 'whether' needs to follow assess
line 206: consistently is not the correct word.  perhaps 'most often' since there are some where others were much more prominent.  typing os too small to read. 

Author Response

If the pathogen detected in a metagenomic survey has already been proved to cause the disease using Koch's postulates mention that. If not, since it can be grown it in pure culture, verify it causes the disease after inoculation and can be recovered from lesions on 'fresh' Camella plants.

Were combinations of two or mare microbes tested for 'combined effects'? that could also be interesting.

Response: We appreciate the reviewer’s valuable comments. As suggested, we have added additional information on pathogenicity tests that were conducted using a single isolate (GNUFP284) of Pestalotiopsis sp., which was detected through metagenomic analysis (Lines 230-235).

Regarding the possibility of combined effects of multiple microbes, we acknowledge this as an important and interesting aspect. However, only single putative pathogen causing leaf spot disease on Camella plants was isolated in this study. We plan to explore potential synergistic or antagonistic interactions among microbes in future research.

Fig 3 Needs larger print and more explanation; I have no idea what the nodes imply. Would a Table
work better\

Response: We thank the reviewer for this important suggestion. To improve clarity, we have revised the figure legend to provide a more detailed explanation of the node structure, color gradient, and symbol meanings. Specifically, the figure presents a cladogram highlighting microbial taxa that are differentially abundant in diseased samples.

Each node represents a taxonomic unit, with the branch color indicating the LDA effect size estimate (red for enrichment and blue for depletion in diseased samples). The size of each node reflects the relative prevalence of the corresponding taxon. Taxa with statistically significant differences (p < 0.05) are indicated with a black outline.While we acknowledge that tabular data can be informative, the figure efficiently conveys both phylogenetic relationships and statistical relevance, which may be less intuitive in tabular form. Therefore, we respectfully chose to retain the figure with an improved legend rather than replace it with a table.

So far as I know, the use of metagenomic analysis for identification of a disease causing organism is unique.  

Editing: line 97,  move per tree to right after leaves
line 144: 'if ' or 'whether' needs to follow assess
line 206: consistently is not the correct word.  perhaps 'most often' since there are some where others were much more prominent.  typing os too small to read. 

Response : We appreciate the reviewer’s insightful comments above; to take suggestions into account, we have changed it as suggested.

Round 2

Reviewer 2 Report

Just need to some improvements  

1. Line 85-89: From "It was hypothesized that" to end:
Delete it or mixing within paragraph before last one in introduction

2. "2.4. Data preparation and statistical analysis":
This section remove it to last section in material and methods

3. The authors must add a separate section for conclusions without repeating the results and discussion

Author Response

1. Line 85-89: From "It was hypothesized that" to end:
Delete it or mixing within paragraph before last one in introduction

Response: Thank you for your suggestion, we have revised it as suggested (It was hypothesized that - deleted)

2. "2.4. Data preparation and statistical analysis":
This section remove it to last section in material and methods

Response: We appreciate the reviewer’s valuable comment. As the experimental design aimed to demonstrate a direct association between the estimated pathogen density (as inferred by metagenomic analysis) and disease occurrence, we further conducted culture-based isolation to confirm the identity of the dominant pathogen detected in the metagenomic data. Therefore, we believe that structuring the manuscript according to each experimental section—metagenomic analysis followed by culture-based validation—is appropriate and scientifically sound. We hope the reviewer agrees with this approach. Thank you once again for your insightful feedback. 

3. The authors must add a separate section for conclusions without repeating the results and discussion

Reponse: Thank you for the valuable comments; As requested, the above content has been added to the end of the manuscript.

Reviewer 4 Report

I feel the paper is now ready for publication; the problems have been corrected to my satisfaction.

I feel the paper is now ready for publication; the problems have been corrected to my satisfaction.

Author Response

I feel the paper is now ready for publication; the problems have been corrected to my satisfaction.

Response: Thank you very much for your valuable comments, which improved our manuscript significantly. 

Round 3

Reviewer 2 Report

No major comments

I repeat my comment. As follows:

1. "2.4. Data preparation and statistical analysis":

This section remove it to last section in material and methods

Author Response

I repeat my comment. As follows:

1. "2.4. Data preparation and statistical analysis":

This section remove it to last section in material and methods

Response: We appreciate the insightful comment above; to take suggestions into account, we have moved the above-mentioned section to the last section in material and methods as suggested.